# Robust Route Planning under Uncertain Pickup Requests for Last-mile Delivery

## ABSTRACT

Empowered by the widespread adoption of Internet of Things (IoT) devices and smartphones, last-mile delivery services have evolved to accommodate both delivery and pickup tasks. An essential challenge in last-mile delivery is efficiently planning routes for couriers to handle pre-scheduled delivery requests as well as stochastic pickup requests. Existing work approaches this problem by either adjusting routes on the fly when new requests arise or preplanning routes based on predicted future pickup requests. However, these methods either compromise the optimality of planned routes or heavily rely on the accuracy of predictions. In this work, we take conformal prediction as an opportunity to address the issue of prediction uncertainty. We design ROPU, a novel courier route planning framework for logistics systems that incorporates conformal prediction into reinforcement learning. Our work advances the existing work from two aspects: (i) Pickup request prediction utilizes spatial-temporal conformal prediction to capture historical pickup request patterns, providing a unified spatial-temporal conformal interval with high confidence (ii) A spatial-temporal attention network assesses location importance from various perspectives and enables the actor to perceive time and integrate the spatial-temporal conformal interval. We implement and evaluate ROPU on one of the largest logistics platforms. Extensive experiment results demonstrate that our method outperforms other state-of-the-art methods with improvements of at least 30.49% in the pickup overdue rate, 25.00% in the delivery overdue rate, and 5.49% in the traveling distance metric.

## CCS CONCEPTS

• **Do Not Use This Code → Generate the Correct Terms for Your Paper**; *Generate the Correct Terms for Your Paper*; Generate the Correct Terms for Your Paper; Generate the Correct Terms for Your Paper.

## KEYWORDS

Route Planning; Last-mile Delivery; Conformal Prediction

**ACM Reference Format:**
Anonymous Author(s). 2018. Robust Route Planning under Uncertain Pickup Requests for Last-mile Delivery. In *Proceedings of Make sure to enter the correct conference title from your rights confirmation emai (Conference acronym 'XX).* ACM, New York, NY, USA, 9 pages. https://doi.org/XXXXXXX.XXXXXXX

## 1 INTRODUCTION

Last-mile delivery, as an emerging Web-of-Things service in logistics systems, transits parcels from local distribution centers or stations to the final consumers. In this process, each courier is responsible for delivering hundreds pieces of parcels to designated areas (e.g., one or many communities) in one day. Empowered by the pervasive Internet of Things (IoT) devices and smartphones, last-mile delivery has integrated an increasing number of sensor devices, such as GPS, RFID, and WiFi [6, 16, 34]. For example, couriers use their smartphones to record real-time locations and report the status of each parcel to web servers when it is picked up and delivered. This integration enhances the efficiency of couriers [17, 31], improving the traceability of parcels [24], and significantly propels the advancement of entire logistics systems.

As a recent trend of expanding the existing last-mile delivery service, pickup services have been integrated into the process [18]. That is, in addition to scheduled delivery, couriers are also responsible for handling an increasing number of on-demand pickup requests (i.e., pickup parcels from customers for delivery), which account for up to a quarter of the total requests. These requests can originate from any location in the communities at any time, and have to be fulfilled within a short time, often as little as one hour [17]. The timely handling of these requests significantly impacts the courier's delivery behavior and route. Ignoring or inadequately addressing these pickup requests can lead to reduced operational efficiency, as demonstrated in our experiments. Therefore, route planning for couriers that considers both delivery and pickup requests has become an essential aspect of efficient last-mile delivery services.

Route planning is a variant of the classic Vehicle Routing Problem (VRP) or Traveling Salesman Problem (TSP). Its primary objective is to find an optimal route that visits all required locations while minimizing travel distance. While TSP and VRP deal with static locations, real-world routing problems are often more intricate, involving dynamic or stochastic requests and real-time location updates [21]. One straightforward approach to addressing these dynamic routing problems is to modify planned routes in real-time when receiving stochastic requests [19, 26, 32]. However, this method can disrupt the optimality of planned routes, leading to detours to accommodate stochastic requests. For instance, it may require returning to previously visited locations to pick up parcels. Alternative methods [15, 29, 35] involve predictive planning. These approaches consider anticipated future requests when devising routes. Nevertheless, the effectiveness of such methods is contingent upon the accuracy of predictions and may fall short when faced with prediction uncertainties.

In this work, we aim to design a robust route planning framework that can effectively handle the inherent uncertainty associated with predicting future stochastic pickup requests. To this end, we take conformal prediction [25] as an opportunity, which provides

prediction intervals with high confidence levels, regardless of what predictor is used. Our focus here is *spatial-temporal conformal prediction*, where we guarantee the actual pick request happens within a small set of predicted locations (i.e., spatial interval) in a certain predicted duration (i.e., temporal interval) at a high confidence level. By incorporating this high-confidence spatial-temporal interval into our route planning process, we can generate routes that consider both delivery and pickup requests with reduced uncertainty. This enhancement significantly improves the efficiency of last-mile delivery services.

Leveraging conformal prediction for courier route planning has the following two key challenges. First, route planning requires spatial-temporal information, including both the 'where' and 'when' aspects of pickup events. Conformal prediction, as traditionally applied, primarily handles one-dimensional outputs, such as object classes (e.g., cat or dog), which cannot capture the inherent connection between the spatial and temporal dimensions. Moreover, real-world route planning must count for geographical factors such as road network, and the spatial-temporal interval can be affected by distribution shifts over time. [8]. The second challenge involves incorporating spatial-temporal conformal prediction into route planning. In particular, we use a Reinforcement Learning(RL)-based planning method considering it is a common method for solving stochastic VRP [19, 21]. Despite RL's applicability, there is currently a lack of a framework that effectively integrates reinforcement learning with spatial-temporal intervals. Specifically, the challenges lie in adequately representing the spatial-temporal interval within the reinforcement learning context and enabling the policy network to account for the potential impact of the spatial-temporal interval.

To address these challenges, we model courier route planning as a Markov Decision Process and design a route planning framework by combining actor-critic reinforcement learning and spatial-temporal conformal prediction, named ROPU. We first design a spatial-temporal unified conformity score to obtain a spatial-temporal interval for pickup request prediction. This design takes road networks into account as a constraint within the spatial interval while also considering distribution shifts over time in the spatial-temporal interval. Then we present irregular the spatial-temporal conformal interval in a 3D space and integrate them into an RL-based routing planning model. Following that, we create a spatial-temporal attention network serving as the actor. This network has the capability to perceive time and integrate the spatial-temporal conformal interval. It considers the importance of different locations from four aspects: temporal, spatial, actual requests, and requests under conformal prediction, ultimately selecting the most suitable next location.

In particular, our main contributions are as follows.

- To our best knowledge, we are the first to approach the courier route planning problem under uncertain pickup requests for last-mile delivery taking the opportunity of conformal prediction.
- We design ROPU, a novel courier route planning framework for last-mile delivery that incorporates conformal prediction into reinforcement learning. This framework incorporates several innovative components: (i) Pickup request prediction utilizes spatial-temporal conformal prediction

to capture historical pickup request patterns, providing a unified spatial-temporal conformal interval with high confidence, while also considering road conditions.; (ii) A spatial-temporal attention network assesses location importance from four perspectives: spatial, temporal, actual request, and predicted request intervals. This network enables the actor to perceive time and integrate the spatial-temporal conformal interval, allowing it to select the most suitable next action (location).

- We implement and evaluate ROPU on one of the largest logistics platforms, i.e., [anonymous company]. We have conducted extensive experiments based on 4-month real-world data, demonstrating that our method outperforms other state-of-the-art methods with improvements of at least 30.49% in the pickup overdue rate, 25.00% in the delivery overdue rate, and 5.49% in the traveling distance metric.

## 2 DEFINITION AND FORMULATION

### 2.1 Definition

*2.1.1* **Delivery Request**. A delivery request is for couriers to deliver parcels to a customer with a specific deadline, such as 3 p.m. or 12 p.m. The delivery requests are generally conformed a certain duration earlier (e.g., 8 hours) than work hours.

*2.1.2* **Pickup Request**. A pickup request is a stochastic request from customers to pick up parcels at a designated location (e.g., the customer's home) within a specified time period (usually within one hour) for delivery.

*2.1.3* **Delivery Station and Delivery Zone**. A delivery station acts as the origin point for last-mile delivery, where courier assembled parcels based on the delivery locations. The covered area of a delivery station is segmented into several delivery zones (e.g., communities), taking into account various real-world factors such as the configuration of road networks and the dispersion of request destinations [31]. Each delivery zone is assigned to one delivery courier, who is responsible for completing all delivery and pickup requests within the zone.

*2.1.4* **Area of Interest(AOI)**. An area of interest (AOI) is a collection of nearby locations (e.g., building) in a delivery zone and each delivery zone has one or more AOIs. In our problem, AOI is the minimum spatial unit that represents delivery and pickup locations.

### 2.2 Problem Formulation

**Courier Route Planning Problem.** Given a delivery zone, a set of delivery requests, and historical pickup requests in this zone, our goal is to find a function to generate an optimal route for the courier in this delivery zone that maximizes the courier's efficiency (i.e., minimize the total traveling distance) and ensure a good customer experience (i.e., maintain a low overdue rate of requests), considering potential future stochastic pickup requests.

In our work, we formulate courier route planning as a Markov Decision Process (MDP). We define an agent for the last-mile delivery center, which is responsible for providing a delivery route for each courier in their delivery zone. Formally, this problem is

characterized by four major components: $\{\mathcal{A}, \mathcal{S}, \mathcal{P}, \mathcal{R}\}$. $\mathcal{A}$ denotes the action space; $\mathcal{S}$ denotes the set of states; $\mathcal{P}$ denotes the state transition; $\mathcal{R}$ is the reward function. We introduce $\mathcal{A}, \mathcal{S}, \mathcal{P}, \mathcal{R}$ in detail as follows.

**Action $\mathcal{A}$.** The action $a_k$ involves selecting the next location (i.e., AOI) the courier should proceed to at decision epoch $k$. These decision epochs represent the moments when the agent determines actions for creating a planned route. The courier then follows this route to fulfill delivery requests and potential pickup requests along the way. After completing a route, we advance to the next decision epoch, denoted as $k + 1$. We use $t(k)$ to represent the time of decision epoch $k$. It is important to note that the time intervals between two decision epochs are non-uniform due to varying action completion times. During the training process, the agent takes action (i.e., chooses a location) at each decision epoch, collectively forming a route. In contrast, during testing, the agent incrementally generates a complete route based on the current state, using a well-trained model. When a new pickup request is received, the agent recalculates the route accordingly.

**State $\mathcal{S}$.** The state at decision epoch $k$ is defined as $s_k = \{t(k), loc_c^k, loc, d_{de}^k, d_p^k, d_{pre}^k\}$, where $t(k)$ is the time of decision epoch $k$. $loc_c^k$ is the current location of the courier at decision epoch $k$. Each location (i.e., AOI) has these information $\{loc, d_{de}^k, d_p^k, d_{pre}^k\}$. The $loc_i$ is the two-dimensional coordinate of AOI $i$. The $d_{de,i}^k, d_{p,i}^k, d_{pre,i}^k$ of AOI $i$ is defined as follows.

- $d_{de,i}^k$ is the delivery request in AOI $i$ at decision epoch $k$, represented by a vector where different dimensions of the vector represent the request within specific time periods (i.e., the deadline of requests is within a certain time period).
- $d_{p,i}^k$ and $d_{pre,i}^k$ are actual pickup request and predicted pickup request in AOI $i$ at decision epoch $k$, which are also represented by a vector, similar to $d_{de,i}^k$. Predicted pickup requests would not be added to the actual pickup requests until occurred.

**State Transition $\mathcal{P}$.** Before entering the next decision epoch $k + 1$, we update the time and three types of requests according to the real-world data (i.e., actual pickup requests that occurred) and the impact of the action (i.e., delivery and pickup requests served). (1) When an actual pickup request occurs between two decision epochs, we need to add this request to the actual pickup request and make reductions in the predicted pickup request. (2) When delivery and pickup requests are served, we reduce the corresponding actual request as well as decrease the associated predicted request. (3) We update $t(k + 1)$ based on $t(k)$, traveling time, and serving time. It is defined as $t(k + 1) = t(k) + Dis^k \times Speed_{tra} + N_{de}^k \times Speed_{de} + N_p^k \times Speed_p$, where $Speed_{tra}, Speed_{de}$, and $Speed_p$ represents the speed of traveling, delivery serving, and pickup serving, which are estimated from historical speed and service time. $N_{de}^k$ and $N_p^k$ represent the number of customers with delivery requests served and customers with pickup requests served during decision epoch $k$, respectively. Then we integrate them to the next state, denoted as $s_{k+1} = \{t(k + 1), loc_c^{k+1}, loc, d_{de}^{k+1}, d_p^{k+1}, d_{pre}^{k+1}\}$.

**Reward Function $\mathcal{R}$.** Given the state $s_k$ and the action $a_k$ at decision epoch $k$, we calculate the reward $r_k$, which comprises the traveling distance and the overdue rate (i.e., indicating delay of fulfilling requests). A shorter traveling distance and a lower overdue rate indicate a higher reward. The reward is defined as

$$r_k = -\frac{1}{2}\lambda_{de}^{\Delta_{de}} \times Dis - \frac{1}{2}\lambda_p^{\Delta_p} \times Dis, \tag{1}$$

where $Dis$ is the traveling distance between two AOIs (i.e., the last selected location and the next selected location). $\Delta_{de}$ and $\Delta_p$ are the overdue rate of delivery requests and pickup requests during this period. $\lambda_{de}$ and $\lambda_p$, greater than 1, are the penalty factors for delivery requests and pickup requests. $\lambda_{de}^{\Delta_{de}}$ and $\lambda_p^{\Delta_p}$ account for the influence of the overdue rate and serve as penalty terms. The larger the overdue rate, the less the corresponding reward.

Given the above definition, we aim to learn a policy to make decisions, which can be represented as

$$P(A|s_0) = \prod_{k=0}^{K} \pi(a_{k+1}|a_k, s_k), \tag{2}$$

where $s_k$ denotes the state at decision epoch $k$, and A denotes the route composed of a series of actions (locations). Our goal is to find an optimal policy $\pi^*$ and maximize the cumulative reward $\mathbb{E}_{a \sim \pi(\cdot|a,s)}[\sum_{k=1}^{\infty} r(a_k, s_k)]$.

## 3 DESIGN

### 3.1 Overview

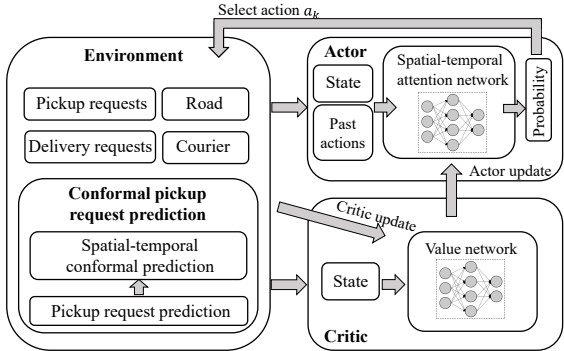

**Fig 1: Framework of ROPU**

We design a reinforcement learning-based courier route planning framework incorporated by conformal prediction, named ROPU, depicted in Fig. 1. There are two modules in ROPU: the environment module and the actor-critic reinforcement learning module. (1) The environment module simulates couriers' last-mile delivery operations based on real-world delivery requests, pickup requests, road networks, and courier status. We further build a pickup request prediction model to predict future pickup requests using historical pickup requests. To quantify the uncertainty of the prediction model, we develop a spatial-temporal conformal prediction method to generate a spatial-temporal interval for each pickup request. (2) The actor-critic reinforcement learning module incorporates real-time observations from the environment to determine the optimal route for a courier. This route is continuously updated to adapt to dynamic changes in requests. Specifically, the agent selects an action as the next location for couriers at each decision epoch and receives a reward from the environment. The action is determined

by an actor, in this case, our designed spatial-temporal attention network. As the state changes, we use the critic to assess the value of the actions made by the actor and provide feedback to it.

In this section, we first introduce conformal pickup request prediction. Then we introduce the spatial-temporal attention network and finally introduce the training method.

## 3.2 Conformal pickup Request Prediction

Considering the significant impacts of the stochastic pickup requests on couriers' routes, we first predict the time and location of pickup requests in a future time period (such as a day) by extract occurring patterns from the historical pickup requests. Then, to quantify the uncertainty of the prediction model, we introduce spatial-temporal conformal prediction.

### 3.2.1 Pickup Request Prediction.
In a delivery zone where multiple pickup requests occur throughout the day, we regard the prediction of pickup requests as a sequence prediction task [9]. Formally, we define the prediction as

$$X_{n+1}, ..., X_{n+K} = f(X_1, ..., X_n), \tag{3}$$

where $X_n = [l_n, t_n]$ denotes the location $l_n$ and time $t_n$ of the $n$-th pickup request. $\{X_1, ..., X_n\}$ represent $n$ historical pickup requests and $\{X_{n+1}, ..., X_{n+K}\}$ represent $K$ future pickup requests. $f$ is a sequence prediction model that can be adjusted using various models, such as LSTM [10] or Transformer [30]. Since the number of pickup requests varies each day, we apply a method borrowed from Natural Language Processing (NLP) [20]. We introduce a start symbol and an end symbol, and then pad any insufficient parts of the sequence with a special symbol. These symbols can be learned gradually during the training process and assigned specific meanings for the sequence's beginning, end, and padding.

### 3.2.2 Spatial-Temporal Conformal Prediction.
Conformal prediction [25] is a technique for quantifying uncertainties of machine learning models. Specifically, given an input, conformal prediction estimates a prediction interval that is guaranteed to cover the true value with high probability. Formally, given a dataset $(X_i, Y_i)_i^n$ and a new observation $(X_{n+1}, Y_{n+1})$, it can be represented as

$$P(Y_{n+1} \in C_\alpha(X_{n+1})) \geq 1 - \alpha, \tag{4}$$

where $Y_{n+1}$ is a true value, $C_\alpha(X_{n+1})$ is the prediction interval, and $\alpha$ is the confidence level. In our problem, we use conformal prediction to generate a spatial-temporal interval for each pickup request, ensuring high confidence.

The general conformal prediction framework [22, 25] is designed for a 1-dimensional value or a 2-dimensional location, which does not capture the inherent connection between the spatial and temporal dimensions. This special requirement motivates us to design a unified spatial-temporal interval to guide courier's routes. Additionally, the spatial-temporal interval needs to consider the constraints of geographical factors such as road networks. The distribution shifts over time can also influence the generation of the spatial-temporal interval [8]. To achieve our goal, we modify the existing conformal prediction framework with the following five steps.

**(i) Identify a score function to quantify inconsistency.** The conformity score is used to evaluate the conformity between the calibration's response values and the predicted values. To measure the conformity of the spatial-temporal outputs, we define it as a vector $m_i = [m_{i,sp}, m_{i,t}, m_{i,spt}]$, where $m_{i,sp}, m_{i,t}, m_{i,spt}$ are spatial, temporal, and spatial-temporal prediction errors, respectively. $m_{i,sp}$ is the distance error, defined as $Dis_g(l_{\hat{y}}, l_y)$, where $l_{\hat{y}}$ is two-dimensional coordinates of the predicted location, $l_y$ is the true location, and $Dis_g$ is the distance between them. $m_{i,t}$ is the temporal error, defined as $|t_{\hat{y}} - t_y|$, where $t_{\hat{y}}$ and $t_y$ are the predicted time and true timem, respectively. $m_{i,spt}$ measures the spatial-temporal joint error, defined as $\sqrt{(u(m_{i,sp})^2 + u(m_{i,t})^2}$, where $u(\cdot)$ represents a normalization operation, as $m_{i,sp}$ and $m_{i,t}$ are not on the same scale.

**(ii) Compute $(1 - \alpha)$-th quantile of scores.** After that, we calculate $(1 - \alpha)$-th quantile of the conformity scores, denoted as $\widehat{Q}_{1-\alpha}(M_{cal})$. This quantile is based on $m_{spt}$ to ensures coverage both in spatial and temporal aspects. Specifically, $\widehat{Q}_{1-\alpha}(M_{cal})$ is composed of three dimensions $[\widehat{Q}_{1-\alpha}^{sp}(M_{cal}), \widehat{Q}_{1-\alpha}^{t}(M_{cal}), \widehat{Q}_{1-\alpha}^{spt}(M_{cal})]$.

**(iii) Construct prediction interval.** Based on $\widehat{Q}_{1-\alpha}(M_{cal})$, we can calculate spatial-temporal prediction interval $C_\alpha(X_{n+1}) = [C_\alpha^{sp}(X_{n+1}), C_\alpha^t(X_{n+1})]$. The spatial interval, $C_\alpha^{sp}(X_{n+1})$, is defined as $[\hat{Y}_{n+1}^{sp} \pm \widehat{Q}_{1-\alpha}^{sp}(M_{cal})]$ and the temporal interval, $C_\alpha^t(X_{n+1})$, is defined as $[\hat{Y}_{n+1}^t \pm \widehat{Q}_{1-\alpha}^t(M_{cal})]$. The resulting spatial-temporal interval we constructed is a 3D geometry in the spatial-temporal space, as shown in Fig. 2.

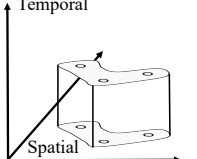

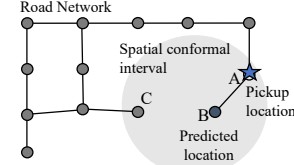

Fig 2: 3D spatial-temporal interval

Fig 3: Impact of geographical factors on route planning.

**(iv) Geographical factors.** We consider the impact of geographical factors on route planning. Fig. 3 illustrates this impact, with A and B representing the actual and predicted pickup request locations, respectively. The light gray circle denotes the prediction interval without considering geographical factors, which includes point C. In reality, due to road separation, C is distant from A and B. Including C in the interval could lead to inaccurate route planning. Therefore, we incorporate a road network and use road distance to calculate $Dis_g$ when computing spatial conformal scores.

**(v) Distribution shifts over time.** For the temporal aspect, the time sequence data do not conform the exchangeable assumption due to distribution shifts [8, 33]. To address this issue, we adopt a technique inspired by [8] for online updates. We first compute the interval based on existing data. Then whenever a new request occurs, we update both the spatial-temporal interval $C_{\alpha_{k+1}}^{k+1}(X_{k+1})$ and the confidence level $\alpha_{k+1}$ after $k$ pickup requests occurring. The update of $\alpha_{k+1}$ is defined as $\alpha_{k+1} = \alpha_k + \beta(\alpha - \mathbb{I}\{y_k \notin C_{\alpha_k}(X_k)\})$, where $\beta > 0$ is a step size parameter and $\alpha$ is the original confidence level. After updating $\alpha_{k+1}$, we also adjust the associated spatial-temporal interval and make appropriate modifications in the next state $s_{k+1}$.

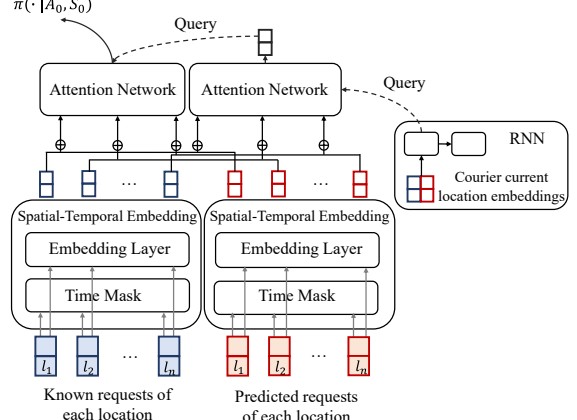

**Fig 4: Spatial-temporal Attention Network**

## 3.3 Spatial-Temporal Attention Network.

In our RL framework, the actor is used to select actions (i.e., next location) based on the probabilities provided by the actor network. Motivated by [19], we utilize an attention network as the actor. To incorporate the time windows and the spatial-temporal interval of predicted pickup requests, we design a spatial-temporal attention network. Fig. 4 shows the structure of the spatial-temporal attention network. (i) The inputs of the network are the known (pickup and delivery) requests and the predicted pickup requests at each location. Then we encode the known and predicted requests for all locations by a spatial-temporal embedding layer. Subsequently, we obtain two kinds of embedding and concatenate them for the same location. (ii) We feed the embedding of the current courier location into an RNN to obtain the hidden state. (iii) Then we use the hidden state from the RNN as the query to obtain attention scores for different locations and get the probability of each location being selected. The spatial-temporal attention network computes the importance of the actual (pickup and delivery) requests and conformal predicted pickup requests separately, in order to endow the actor with the ability to perceive time and integrate the spatial-temporal conformal interval.

*3.3.1 **Spatial-Temporal Embeddings**.* Let $h_i^k = (loc_i, d_{de,i}^k, d_{p,i}^k)$ be the feature of location $i$ to represent the known requests, where $loc_i, d_{de,i}^k, d_{p,i}^k$ is the location coordinate, the delivery request, and the actual pickup request of location $i$, respectively. These features are be directly fed into the embedding layer to obtain the spatial dimension embedding $\bar{h}_{i,sp}^k$. Additionally, we add a time mask on the feature in the time dimension before feeding it into the embedding layer to obtain the time dimension embedding $\bar{h}_{i,te}^k$. The time mask here is used to hide the requests from previous time steps. We concatenate the embeddings of spatial and temporal dimensions to obtain $\bar{h}_{i,kn}^k = [\bar{h}_{i,sp}^k \oplus \bar{h}_{i,te}^k]$, where $\oplus$ denotes concatenation.

In addition, we let $h_{i,pre}^k = (loc_i, d_{pre,i}^k)$ be the feature of location $i$ to represent the predicted requests. We apply the same embedding and time mask operations to this feature, producing embeddings in both the spatial and temporal dimensions. Notably, the time mask here conceals requests from previous time steps and from the next

few hours, effectively filtering out requests that cannot be fulfilled in the near future. Then we obtain concatenated embeddings $\bar{h}_{i,pre}^k$. Finally, by concatenating the embeddings of the known requests and predicted requests for each location, we obtain $\bar{h}_i^k = [\bar{h}_{i,kn}^k \oplus \bar{h}_{i,pre}^k]$.

*3.3.2 **Attention Network**.* We employ an RNN for the decoder and the input of the RNN is the embedding of the courier's current location. At each decoder step, we use the hidden state of the RNN as the query to calculate the importance of each location for the next decoding step. Let $z^k \in \mathcal{R}^D$ represent the hidden state of the RNN cell at decoding step $k$. We compute the attention scores as $\alpha^k = softmax(u^k)$, where $u_i^k$ is defined as $u_i^k = \sigma(W_a[\bar{h}_i^k \oplus z^k])$. Here, $W_a$ is trainable weight, and $\sigma$ is the activation function. Then we calculate the weighted sum of the vector, defined as $c^k = \sum_i^M \alpha_i^k \bar{x}_i^k$. We calculate the conditional probabilities by combining the context vector $c_k$, computed as $\chi_i^k = v_c\sigma(W_c[\bar{x}_i^k \oplus c^k])$, where $v_c$ is a trainable variable. Finally, we normalize the values using a softmax function to obtain the selected probability

$$\pi(|Y_k, X_k) = softmax(\chi^k). \tag{5}$$

For generating a feasible route, we use a masking scheme that sets the log probabilities of infeasible locations to $-\infty$. Specifically, we use the following masking procedures: (i) spatial mask: locations with no requests are not allowed to be visited; (ii) temporal mask: locations cannot be accessed if the requests' deadlines are $n$ hours later than the current time.

## 3.4 Training Method

In our RL framework, the critic evaluates the action and provides feedback to the actor. The critic serves as a function estimator, taking the agent's state as input and outputting the state value. The state value function, denoted as $V_{\theta_c}(s_k)$, represents the expected long-term value at decision epoch $k$ under state $s_k$.

Then we introduce the updated strategy of the actor and critic. Firstly, we update the weights of the critic using time difference methods [2]. The network parameters $\theta_c$ are updated by minimizing the following loss function

$$L_{\theta_c} = \frac{1}{2}[r^{k+1} + V_{\theta_c}(s_k) - V_{\theta_c}(s_{k+1})]^2, \tag{6}$$

where $\theta_c$ is the parameters of the value network. Secondly, we update the weights of the actor based on an advantage function

$$A(s^k, a^k) = r^{k+1} + V_{\theta_c}(s_k) - V_{\theta_c}(s_{k+1}), \tag{7}$$

where the advantage function is used to reduce the high variance of the policy networks and stabilize the model. With the advantage function, we define the gradient of actor by

$$\nabla_{\theta_\rho} J(\theta) = \nabla_{\theta_\rho} \pi_{\theta_\rho}(s^k, a^k) A(s^k, a^k), \tag{8}$$

where $\theta_\rho$ is the weight of actor and $\pi_{\theta_\rho}(s^k, a^k)$ is the policy probability function. Then we update the actor's parameters $\theta_\rho$ by the gradient descent rule as $\theta_\rho \leftarrow \theta_\rho + \eta\nabla_{\theta_\rho}J(\theta)$, where $\eta$ is the learning rate.

# 4 EVALUATION

In this section, we first introduce our datasets, followed by the experimental settings. Then, we introduce the performance of our method.

## 4.1 Dataset Description

We conduct experiments on a real-world dataset, collected from [anonymous company], one of the largest logistics companies in China. The dataset includes delivery and pickup parcel information, as well as courier trajectories in the central area of a city. The parcel information includes details such as location, start time, deadline, and finish time, while the trajectories contain latitude and longitude information. We focus on a 4-month period of delivery service data, which included over 960,000 pickup requests and 290,000 delivery requests. An example of the data is provided in Table 1.

**Table 1: An example of data.**

| Pickup info | Location | Start time | Deadline | Finish time |
|---|---|---|---|---|
| | A001 | 01-12 10:30:30 | 01-12 11:30:30 | 01-12 11:15:02 |
| Delivery info | Location | Start time | Deadline | Finish time |
| | A002 | 01-12 06:17:30 | 01-12 15:00:00 | 01-12 10:12:00 |
| Trajectory data | Id | Longitude | Latitude | Timestamp |
| | D001 | 121.39 | 37.50 | 01-12 09:00:30 |

## 4.2 Experimental Setup

*4.2.1 **Evaluation Configuration**.* For the time settings, we divide the 24 hours of a day into 15-minute intervals, resulting in a total of 96 time slots. For the spatial setting, we aggregate the locations of all delivery and pickup requests within the same AOI, using the central coordinates of the AOIs as the delivery and pickup request locations. Additionally, we leverage the road network to obtain the network distance between AOIs, which is employed in determining the conformity score and the courier's travel distance.

For conformal pickup request prediction, we train the model using data from the first two months and test it with data from the last month. During training, we utilize pickup request records from the previous 7 days to predict the pickup requests for the next day. We set a maximum of 100 pickup requests per day based on historical records. For the route planning, the courier's speed is set at 25 km/h, and the serving time for a customer with a delivery request is 1 minute, while the serving time for a customer with a pickup request is 10 minutes, based on the analysis of couriers' trajectories.

*4.2.2 **Implementation**.* We implement our method and baselines with tensorflow 1.4.0 in Python 3.6 environment and train it with 16GB memory and Intel(R) Xeon(R) CPU E5-2680 v4 @ 2.40GHz (CPU). We apply the Adam optimizer with a learning rate of 1e-4. We set the embedding size in actor and critic as 256. We set the penalty factor of delivery and pickup as 1.5 and 2, respectively.

*4.2.3 **Baselines**.* We compare our model with the following baselines and variants of our model:

- **No-routing**. This is the real-world data about how couriers decide their route by themselves.

- **NCO [3]**. This is a framework tackling combinatorial optimization problems using neural networks and reinforcement learning.
- **RL-VRP [19]**. This is a reinforcement learning framework for solving vehicle routing problems. Considering our scenario includes uncertain pickup requests, we adopt an actor-critic training strategy.
- **DRLSA [11]**. It is designed for stochastic vehicle routing problems. They combine neural networks-based Time Difference (TD) learning with experience replay to approximate value functions.
- **MC-VRP [32]**. MC-VRP is an online approach to solve the dynamic vehicle routing problem. Considering the different settings, we remove the Monte Carlo tree search.

Variants of our model

- **ROPU without Conformal Pickup Request Prediction (w/o CPRP)**. In this setting, we remove pickup request prediction. We recalculate the optimal route each time when a new pickup request occurs.
- **ROPU without Conformal Prediction (w/o CP)**. In this setting, we remove the conformal prediction and directly use the results from the prediction.
- **ROPU without Predicted Request Attention (w/o PRA)**. In this setting, we remove the part designed for predicted requests in the spatial-temporal attention network.
- **ROPU without Spatial-Temporal Attention Network (w/o STAN)**. In this setting, we use a general attention network to replace the spatial-temporal attention network.

*4.2.4 **Metric**.* The evaluation metrics are as follows.

- **Traveling distance**: traveling distance represents the total distance a courier covers to fulfill all pickup and delivery tasks, with shorter distances indicating higher efficiency.
- **Overdue rate**: for $N$ delivery tasks, the overdue rate is defined as $\frac{N_{de}^O}{N}$, where $N_{de}^O$ represents the number of delivery tasks that are actually completed after their respective deadlines. A lower overdue rate indicates a better customer experience. For pickup tasks, we define the overdue rate in the same manner. It should be mentioned that $N_p^O$ represents the number of delivery tasks that are actually completed after their deadlines and 2 hours before their start time. We take the start time into account because picking up too early can lead to a negative experience for the customer.

## 4.3 Overall Performance

We compare our approach with the baselines, and the comparison results are in Table 2. From the results, we have following findings: (1) Our method outperforms other baselines in all the performance metrics. In summary, our method has achieved at least a 30.49%, 25.00%, and 5.49% improvement in the metrics of pickup overdue rate, delivery overdue rate, and traveling distance, respectively. Our approach integrates the spatial-temporal interval of future pickup requests into reinforcement learning-based route planning, which equips the courier with the capability to handle potential stochastic pickup requests.

**Table 2: Overall Performance. Bold scores are for the best values.**

| Method | Pickup overdue rate | Delivery overdue rate | Traveling distance (km) |
|---|---|---|---|
| No routing | 0.212 | 0.085 | 35.426 |
| NCO | 0.196 ± 0.022 | 0.125 ± 0.006 | 35.013 ± 1.006 |
| RL-VRP | 0.185 ± 0.007 | 0.086 ± 0.007 | 33.180 ± 0.842 |
| DRLSA | 0.164 ± 0.009 | 0.076 ± 0.011 | 32.452 ± 0.927 |
| MC-VRP | 0.173 ± 0.007 | 0.077 ± 0.006 | 33.006 ± 1.357 |
| **ROPU** | **0.114 ± 0.008** | **0.057 ± 0.004** | **30.671± 0.503** |

(2) Compared to real-world scenarios (i.e., no-routing), our method has improved the delivery and pickup overdue rate by 32.94% and 46.22% and reduced the courier's traveling distance by 13.42%. This indicates that our approach can help logistics platforms enhance courier efficiency and improve the customer experience.

(3) Compared to methods that are not designed for stochastic pickup requests (i.e., NCO), those designed for stochastic requests perform better. We can observe that NCO sacrifice some delivery request performance to compensate stochastic pickup requests. In addition, all methods have a decreased overdue rate for pickup requests.

## 4.4 Results of Pickup Requests Conformal Prediction

*4.4.1 Pickup Request Prediction Results.* For pickup request prediction, we choose three common prediction models, i.e., XGBoost [4], LSTM, and Transformer. We treat this prediction problem as a classification problem. We first obtain the accuracy of location and time. Then we can calculate the absolute error of distance and time. The results are shown in Fig. 3. As evident from the figure, using these models directly for pickup request prediction leads to significant errors in both time and location. Using such inaccurate results for route planning would adversely affect routing performance.

**Table 3: Pickup Request Prediction Performance**

| Method | Accuracy of location | Absolute error of distance (meter) | Accuracy of time | Absolute error of time (minute) |
|---|---|---|---|---|
| XGBoost | 0.457 | 231.765 | 0.486 | 59.236 |
| LSTM | 0.112 | 478.633 | 0.135 | 112.312 |
| Transformer | 0.576 | 166.723 | 0.637 | 42.761 |

*4.4.2 Conformal Prediction Results.* To quantify the uncertainty of the predictions, we employ conformal prediction on the pickup request prediction results. Specifically, we select the transformer with the best performance and apply spatial-temporal conformal prediction to guarantee the accuracy of its prediction results. When we set the confidence level as 0.1, we can achieve 89.8% coverage. That is, 89.8% of the cases, the true location and time of the pickup requests are within our prediction interval. The mean length of the spatial prediction interval is 372 meters and the mean length of the temporal prediction interval is 85 minutes.

## 4.5 Ablation Study

*4.5.1 The Effect of the Conformal Pickup Request Prediction.* To assess the significance of conformal pickup request prediction,

we conducted a comparison among our model ROPU and its two variants (i.e., w/o CP and w/o CPRP). The performance of ROPU and its variants is displayed in Fig. 5. We first analyze the importance of conformal prediction. We observed that when the predicted results guide the courier's route planning (i.e., ROPU w/o CP), the results are worse than those achieved with ROPU. This suggests that inaccurate prediction results can indeed impact route planning. We further investigate the importance of conformal pickup request prediction. In this setting, we neither predict future pickup requests nor employ conformal prediction. We observe that ROPU w/o CPRP performs worse than both ROPU and ROPU w/o CP. The possible reason is that, without prior knowledge of the pickup requests, the courier's original route may conflict with the pickup requests, both spatially and temporally. As a result, the couriers face challenges in balancing delivery efficiency and service experience.

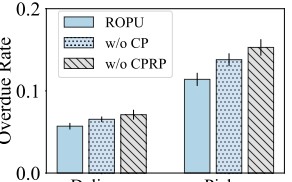
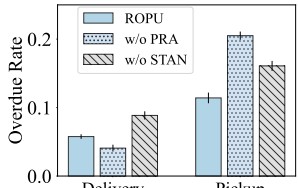

**Fig 5: The Effect of the Conformal pickup Request prediction**

**Fig 6: The Effect of Spatial-Temporal Attention Network**

*4.5.2 The Effect of the Spatial-Temporal Attention Network.* To understand the importance of the spatial-temporal attention network, we compare our model ROPU with its two variants (i.e., w/o PRA and w/o STAN). Firstly, we remove the part designed for predicted requests in the spatial-temporal attention network. We observe that the pickup overdue rate of ROPU w/o PRA increases and its delivery overdue rate decreases. The possible reason is that after removing the spatial-temporal attention network part that focuses on the predicted requests, the model tends to prioritize the delivery tasks, which in turn shows the effectiveness of the spatial-temporal attention network in integrating conformal prediction. Next, we replace the spatial-temporal attention network with a standard attention network. We observe that the performance deteriorates, and the overdue rate of both delivery and pickup increase. It indicates that our designed spatial-temporal attention network is suitable for considering spatial-temporal intervals for route planning.

## 4.6 Impact of Factors

*4.6.1 The Impact of the Ratio of Pickup Requests.* In real-world scenarios, the quantity of stochastic pickup requests varies. To explore our model's performance under different conditions, we manipulated the number of pickup requests, setting them at 0%, 35%, 70%, and 100% of the actual quantity, as depicted in Fig. 7. Observing the figure, it becomes apparent that as the number of stochastic pickup requests increases, the model is required to handle a greater volume of these requests, resulting in higher overdue rates for both delivery and pickup requests.

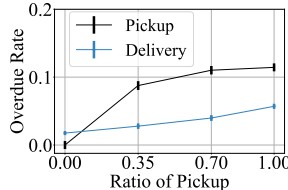

**Fig 7: The impact of the ratio of pickup requests**

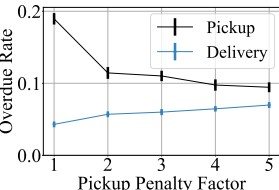

**Fig 8: The impact of the penalty factor**

*4.6.2* ***The Impact of the Penalty Factor.*** The penalty factor is an essential parameter in the reward function as it determines the relationship between the traveling distance and the overdue rates for pickup and delivery tasks. To assess our model's performance under various penalty factors, we keep the delivery penalty factor fixed at 1.5 while adjusting the pickup penalty factor, as shown in Fig. 8. We see that as the penalty factor for pickup increases, our model pays more attention to the overdue issues of pickup requests. As a result, we can achieve a lower overdue rate for pickup. However, this comes at the cost of increased travel distance and overdue rates for deliveries. To strike a balance among these metrics, we select the pickup penalty factor of 2.

## 5 DISCUSSION

**Lessons learned:** Based on the results from our paper, we summarize the following lessons learned:

- Conformal prediction provides a high-confidence spatial-temporal interval for predicting uncertain pickup events, which can guide the courier's route. As shown in Fig. 5, when we only use the predicted results to guide the courier's route planning, the result is worse than considering conformal prediction.
- Spatial-temporal attention network as the actor has the ability to perceive time and integrate the spatial-temporal conformal interval. As shown in Fig. 6, the performance deteriorates (i.e., the overdue rate increases) when the spatial-temporal attention network is removed.

**Limitation:** (i) We evaluate our model on a real-world dataset from one of the largest logistics companies in China. However, due to data privacy and sensitivity concerns, the evaluation dataset is only in the center area of Beijing and last 4 month. We further expect to analyze our model with the dataset covering multiple cities and longer durations. (ii) In this work, we only consider planning routes for couriers without taking into account their acceptance level. This aspect will be considered in our future work.

**Ethics and privacy:** The delivery and pickup requests data and couriers' trajectory data utilized in this work are recorded by the [anonymous company] platform in order to provide better services to its users. For delivery and pickup requests, no specific customer information is involved here; the only data we utilize is location information. However, in this work, we substitute the specific location with an Area of Interest (AOI) ID, thereby ensuring that user privacy is not compromised. Couriers' trajectory data is under the consent agreement of the couriers, and we do not utilize this information to track the detailed trace of the couriers but only infer delivery time.

## 6 RELATED WORK

### 6.1 Route Planning

Route planning is a variant of the classic Vehicle Routing Problem (VRP) [13] or Traveling Salesman Problem (TSP) [12], which aims to find a route to meet all locations' requests with a minimal travel distance. The Vehicle Routing Problem (VRP) can be categorized into: the Standard Vehicle Routing Problem and the Dynamic/Stochastic Vehicle Routing Problem. For the Standard Vehicle Routing Problem, the demands for all locations are known beforehand. Some methods [3, 5, 14, 37] utilize optimization or reinforcement learning based methods to obtain the optimal routes.

However, the actual situation is complex, and some customers' requests may be generated randomly [21]. To handle such situations, some methods [19, 26, 32] either recalculate the route when new requests arise or continually modify it to accommodate new orders. These methods, however, can potentially disrupt pre-existing plans and may result in additional detours due to conflicts between delivery and pickup requests. Some methods [7, 15, 29, 35], when conducting route planning or making recommendations, attempt to predict or estimate future demands in advance. However, the effect of these methods is heavily contingent on the accuracy of the prediction for future demand because an inaccurate estimation can significantly impact performance.

### 6.2 Conformal Prediction

Conformal Prediction aims to provide a prediction interval accompanied by associated confidence levels. It does not impose strong assumptions about the data distribution; it simply requires the data to satisfy the exchangeability assumption [25, 28]. Some methods [1, 23] focus on classification problems. Given an input, the conformal provides a confidence interval or probability set for each possible classification label. Some methods [22, 25] aim to build a conformal prediction for regression problems. Conformal prediction provides a prediction interval instead of a single point estimation. This interval can offer insights into the uncertainty or variability of the predicted value. Some methods [8, 27, 33, 36] aim to design a conformal prediction framework for time-series forecasting tasks, which don't meet the exchangeable assumption. For example, [8] achieves an adaptive conformal inference by treating the distribution shift as a learning problem, where the optimal value of a single parameter changes over time and requires constant re-estimation.

However, these methods are not suitable for our scenario because the general conformal prediction does not meet the requirements of route planning, nor is there a specific framework to integrate conformal prediction into reinforcement learning.

## 7 CONCLUSION

In this work, we focus on robust route planning under uncertain pickup requests for last-mile delivery. We design a novel courier route planning framework called ROPU, which combines conformal prediction with reinforcement learning. ROPU consists of two key components: (i) pickup request prediction with spatial-temporal conformal prediction and (ii) a spatial-temporal attention network with the ability to integrate the spatial-temporal conformal interval. Extensive experiments demonstrate the effectiveness of our model.

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
