# OpenReview forum: "Robust Route Planning under Uncertain Pickup Requests for Last-mile Delivery"
_ACM.org/TheWebConf/2024/Conference — TheWebConf24_

### Official Review · Reviewer_CSin · 2023-11-15

**Novelty:** 5
**Technical Quality:** 5

**Review:**

ADVANTAGE
①	The author chooses abundant baseline methods, such as NCO, RL-VRP, DRLSA and MC-VRP.
②	The experiment includes ablation experiments, enhancing the reliability of the results.
③	The experiment provides a detailed explanation of the ROPU framework, which is advantageous for readers in understanding the model.
④	The article provides a comprehensive summary of lessons learned in the discussion section, contributing to a more cohesive structure and increased readability of the paper.

WEAKNESS
①	The dataset covers a limited number of cities and has a relatively short time span. The article only relies on the dataset from a logistics company in the central area of a city in China, spanning only four months. It would be beneficial if the dataset encompassed cities with diverse geographical locations and terrains, had an extended time frame, or was divided into quarters."
②	The experiment does not account for variations in the proficiency levels of couriers, raising concerns about the practical applicability of the model in real-world scenarios.
③	The experimental evaluation metrics only include overdue rate and travel distance. Including additional metrics such as delivery accuracy, speed, courier job satisfaction, and cost-effectiveness would provide a more comprehensive assessment of the strengths or weaknesses of the model.
④	The paper provides no pseudocode of the experiment, which diminishes the reproducibility.
⑤	The experiment dose not explore parameter sensitivity. Conducting comparative experiments with varying parameters would enhance the reliability of the results.

**Questions:**

How is the mean calculated for spatial prediction intervals with a length of 372 meters and temporal prediction intervals with a length of 85 minutes? Are there significant variations in interval lengths, either larger or smaller? If so, what is the reason behind these variations?
Has the accuracy of the predictions improved by employing a space-time collaborative approach? Could you provide detailed results on this improvement? How does the accuracy compare to using traditional methods?

**Reviewer Confidence:**

2: The reviewer is willing to defend the evaluation, but it is likely that the reviewer did not understand parts of the paper

**Scope:**

2: The connection to the Web is incidental, e.g., use of Web data or API

---

### Official Review · Reviewer_QJWy · 2023-11-16

**Novelty:** 4
**Technical Quality:** 4

**Review:**

This paper proposes ROPU, a courier route planning framework considering the uncertainty of pickup requests. The core of the solution uses a spatio-temporal attention network to predict the pickup requests by exploiting the spatial and temporal characteristics, and then uses a reinforcement learning framework (actor critic network) to do the route planning. The authors have evaluated their solution with the dataset from one of the largest logistic companies in China.

**Quality --**

The overall quality of the paper is good. The authors have framed the problem as an optimization under uncertainty, justified the use of reinforcement learning and spatio-temporal embedding with attention, and then discusses the solution and evaluation results.

**Clarity --**

The overall writing of the paper is satisfactory. It is easy to follow and understand their solution approach.

**Originality --**

I will rate the originality of this paper as low. Use of actor critic network along with spatio-temporal attention for such kind of decision problems have been well studied in the literature. Please check the following literature.

[a] Shen, Bilong, et al. "Stepdeep: A novel spatial-temporal mobility event prediction framework based on deep neural network." Proceedings of the 24th ACM SIGKDD international conference on knowledge discovery & data mining. 2018.

[b] Liang, Yuebing, and Zhan Zhao. "Nettraj: A network-based vehicle trajectory prediction model with directional representation and spatiotemporal attention mechanisms." IEEE Transactions on Intelligent Transportation Systems 23.9 (2021): 14470-14481.

[c] Zhou, Fan, et al. "Semi-supervised trajectory understanding with poi attention for end-to-end trip recommendation." ACM Transactions on Spatial Algorithms and Systems (TSAS) 6.2 (2020): 1-25.

[d] Huang, Liwei, et al. "An attention-based spatiotemporal LSTM network for next POI recommendation." IEEE Transactions on Services Computing 14.6 (2019): 1585-1597.

[e] Huang, Huiqun, Xi Yang, and Suining He. "Multi-Head Spatio-Temporal Attention Mechanism for Urban Anomaly Event Prediction." Proceedings of the ACM on Interactive, Mobile, Wearable and Ubiquitous Technologies 5.3 (2021): 1-21.

[f] Yang, Xi, et al. "Spatio-temporal graph attention embedding for joint crowd flow and transition predictions: A wi-fi-based mobility case study." Proceedings of the ACM on Interactive, Mobile, Wearable and Ubiquitous Technologies 5.4 (2021): 1-24.

Although the problem space addressed in the above papers are little different from the problem studied in this paper, all of the above papers addresses some notion of spatio-temporal decision making under uncertainty, and the corresponding solutions have been modelled as some form of spatio-temporal deep learning frameworks with attention-based embedding. Therefore, the originality of the methodology is little low. I would rather rate this paper on the aspect of novelty in the problem statement that has been solved with a well-explored modelling technique from the spatio-temporal domain.

** Significance -- **

I am doubtful about the significance of this work in the Mobile, IoT track of the Web conference, although the paper is somewhat related to the web. I think the paper fits better for the conferences like ACM SIGSPATIAL or ACM KDD.

** Review Summary **

Here I summarize the pros and cons of the paper.

**Pros:**

+ The problem statement is well motivated.

+ The authors have provided a working solution by adopting a well-explored technique.

+ The evaluation is thorough and shows the importance of the proposed solution.

**Cons:**

- The paper has low significance on IoT, mobile and wearables.

- The solution approach has been well-studied in the context of spatio-temporal decision problems with uncertainty

**Questions:**

1. The problem considers the uncertainty of pick-up requests, but to have the route planning, the uncertainty in the road network should also be considered, for example, the impact of traffic congestion. Can the proposed solution handle that? Does the dataset considered for evaluation consider such kind of road uncertainty?

2. The authors mentioned that the training has been performed on two months data, whereas the testing has been performed with one month data. This captures the temporal variation, however, may not capture the spatial variation. What would be the evaluation results if the model is trained with the one set of locations and tested on different locations which are not been observed on the training data?

In addition, it would be good if the author clarifies about the significance and originality aspects that I have highlighted in my above comments.

**Reviewer Confidence:**

4: The reviewer is certain that the evaluation is correct and very familiar with the relevant literature

**Scope:**

2: The connection to the Web is incidental, e.g., use of Web data or API

---

### Official Review · Reviewer_ke1A · 2023-11-20

**Novelty:** 3
**Technical Quality:** 3

**Review:**

### Major issues addressed in the paper:

This paper presents a method that provides optimized route planning for courier services. In addition to the delivery of parcels, part of the service is that parcels can also be picked up on request. These requests can occur at short notice, so that the route planning must be updated in order to a) drive as few routes as possible, b) meet the deadlines for delivery and c) meet the deadlines for collection. In addition to the classic optimization by a spatial-temporal neural network, possible requests for pick-up are estimated by prediction. The model is evaluated with a data set of real deliveries and pick-ups. The efficiency of the routes can be increased compared to other methods (or even none or ad-hoc methods). The influence of the prediction of possible pick-ups is also evident.

### Detailed comments:

The topic of the paper is definitely justified, but the question is how it fits into the scope of the TheWebConf conference. I find the proximity to the “Web-of-Things” mentioned in the introduction a bit lacking.

What actually makes the process robust? It is mentioned in the text, it is in the title, but the paper itself does not address why this procedure would now be robust.

In my opinion, you have to be very careful with statements like (page 2, 165 ff) "To our best knowledge, we are the first to approach the courier route planning problem under uncertain pickup requests for last-mile delivery taking the opportunity of conformal prediction.", especially because in the following points it is said that the comparison is compared to state of the art. So it can't be entirely new. Neither is ad-hoc route planning and the problem outlined here is a variation of it.

The approach itself is clearly described and technically comprehensible. There is one thing I didn't read or missed. Nowhere is there an allocation between individual courier drivers. Rather, the process acts as a global optimization of a single courier trip. Especially in urban scenarios, there isn't just one driver on the road and their position and workload also play a role. So how are new requests assigned to drivers?

In 4.6.1, the pickup request is artificially specified. Here I asked myself why this happens when you have a real data set and could otherwise take real fluctuations into account.

The general processing overhead remains unclear. Is the performance enough to keep to the epoch (here 15 minutes)? How does the process scale?

A small side note: The paper is motivated by the fact that parcel carriers accept ad hoc requests for pickups. If you look at the data set, there are 3x as many pickups as deliveries. Shouldn't the question be reversed? How do I integrate deliveries into the pickup schedule? Although it would certainly not change the procedure itself... still interesting data.

**Questions:**

Why does the approach ROPU hase Robust in the title? What makes the process robust compared to others?

Why is there no collaborative approach and no allocation between different courier drivers?

Why is this paper relevant to The WebConf?

What is the overhead of the process or how does it scale?

**Ethics Review Description:**

The ethical issue with data trace is also adressed by the authors, however, only anonymized data have been used for the study.

**Reviewer Confidence:**

2: The reviewer is willing to defend the evaluation, but it is likely that the reviewer did not understand parts of the paper

**Scope:**

1: The work is irrelevant to the Web

---

### Official Review · Reviewer_xoaC · 2023-11-24

**Novelty:** 5
**Technical Quality:** 5

**Review:**

This paper proposes a ROPU, which is a route planning framework, for last-mile delivery. It incorporates conformal prediction into reinforcement learning. As shown in the evaluation section, the ROPU outperforms other methods in pickup overdue rate, delivery overdue rate, and traveling distance. This paper is somewhat novel, however, the paper is poorly organized. It has poor expressions, in terms of grammar, typos, and formatting. Besides, the datasets are private real-world data and no empirical validations on public data or synthetic datasets, making it hard to reproduce the results.

**Questions:**

1. On page 1, the authors introduce a new carrier route planning framework named ROPU. It's essential to clarify the meaning of the acronym ROPU.
2. Page 1 lacks CCS CONCEPTS generation.
3. The abstract claims ROPU was evaluated on a major logistics platform, but fails to specify which, undermining the claim's credibility.
4. Although page 2 mentions use of data from an anonymous company, scientific papers should provide either a public version of the data or synthetic datasets to ensure method validation and result reproducibility.
5. Page 2, line 200, needs clarification on the term "stochastic request."
6. The notations on page 3 are disorganized and unclear. A notation table is recommended.
7. Page 3, line 255, contains unclear phrasing; a grammar check is needed.
8. On page 3, line 264, separating actual and predicted pickup requests into distinct points would enhance clarity.
9. Page 3, line 279, introduces "Dis" without prior definition.
10. Equation (1) should explain why the reward function is defined negatively.
11. Page 3, line 304, requires clarification on what "learn a policy" entails.
12. The connection between Equation (2) and the optimal policy and cumulative reward needs clarification.
13. Figure 1 should clearly indicate what "critic update" refers to.
14. Page 3, line 339, lacks clarity on how the spatial-temporal conformal prediction method quantifies prediction model uncertainty.
15. It's unclear if Equation (4) on page 4 relates to the policy defined in Equation (2).
16. Page 4, line 415, has an extra left bracket.
17. Section 3.2.2's equation formatting needs correction; change $(M_{cal)} $ to $(M_{cal}) $.
18. The paper does not clearly differentiate the proposed method's novelty from baseline methods and omits non-trivial references.
19. The "Evaluation" section should state the number of repeat experiments conducted for the results in Table 2.
20. The "Evaluation" section lacks clarity on used parameters. Figures 7 and 8 imply the importance of pickup request ratios and penalty factors, questioning the fairness of claiming superior performance for the proposed method.

**Reviewer Confidence:**

4: The reviewer is certain that the evaluation is correct and very familiar with the relevant literature

**Scope:**

3: The work is somewhat relevant to the Web and to the track, and is of narrow interest to a sub-community

---

### Official Review · Reviewer_HSCs · 2023-11-26

**Novelty:** 5
**Technical Quality:** 5

**Review:**

In this work, the authors focus on the route planning problem for last-mile delivery with uncertain pickup requests. They design a courier route planning framework called ROPU, which first conducts pickup request prediction with spatial-temporal conformal prediction to generate the prediction interval about possible locations where future pickup requests may appear, and then leverage an actor-critic RL framework to make real-time routing decisions based on both the observed pickup/delivery requests and the predicted uncertain future pickup requests in different locations.

Pros:
-	This paper considers the route planning problem for last-mile delivery with uncertain pickup requests, which is an important real-world industry problem. Properly solving the problem can save huge costs for the industry.
-	The method authors used includes RL and conformal prediction, which is appropriate for this scenario.
-	The experiment results prove the superiority of the proposed methods compared with other methods not considering the uncertain pickup requests.
Cons:
-	The RL algorithm is not well-described. The design of the state and the reward could be further discussed and improved. The training process of the RL agent needs further clarification (See questions about RL).
-	Some important details about Conformal pickup Request Prediction are missing, and the writing is a little bit confusing in section 3.2 (See questions about Conformal pickup request prediction).
-	More intuition about the design of the spatial-temporal attention network should be given, especially the time mask mechanism and the RNN applied to the courier location sequence. The notations are not explained properly (See questions about Spatial-Temporal Attention Network).

**Questions:**

I have the following questions about the proposed ROPU method:

About RL
1.	The action at each time period is setting the next targeted location, while I wonder whether the route between two locations (a_t and a_(t+1)) can be changed from time to time according to the real-time road status.
2.	The state during agent decision explained in Sec 2.2  is mainly about time, current location, and the information of all potential AOIs (mainly the pickup/delivery requests), while it does not include other factors about the road status and the courier (i.e. the number of the collected parcels) as mentioned in Fig 1. I wonder whether these factors may affect the decision of the RL agent. For example, a bad road status can affect the travel speed and further the overdue, and a courier may not be able to carry too many parcels at the same time.
3.	The design of the reward function needs further explanation. As mentioned in the paper, the immediate reward of each time period mainly considers the travel distance and the overdue penalty within the time period, while I wonder whether the overdue penalization should be considered on a larger scale, i.e., the whole day, instead of each time period. For example, the immediate reward in each time period can be described as r_t=-distance, and the total reward can be defined as R=(\sigma r_t)*(\lambda_(de)^\Delta_(de)+ \lambda_(p)^\Delta_(p).
4.	In Fig 1, the actor considers both the state and the past actions, while the critic considers only the state. According to the Markov property, the main-stream actor-critic methods only consider the current state of the actor, i.e., a_t=\pi(s_t).
5.	The training process in Sec 3.4 is confusing for me. The output of \pi is a probability distribution over all possible actions, and thus it should take the expectation over all possible actions in equation (8). Furthermore, I wonder whether it should be \pi(a^k|s^k) in equation (8) instead of \pi(s^k,a^k). I also wonder why not use some existing methods like PPO to train the RL agents.

About Conformal pickup request prediction:
1.	The writing of this part should be further improved, and currently, it is hard to understand the process of conformal pickup request prediction. Based on my understanding, it first uses pickup request prediction techniques to predict a set of possible pickup request centers (Y) (in 3D temporal-spatial space). Then, for each pickup request center, it calculate the conformity of different value of intervals (quantile Q) and choose the interval that satisfy the 1-\alpha conformity level (I wonder whether it uses grid search to calculate the conformity levels of all potential intervals and then choose the quantile or use end-to-end methods to directly give an interval Q based on the given alpha, and how the conformity levels are calculated) Based on the center and the interval, it generates the prediction interval (C=Y+-Q). In this process, I suggest that the authors give more details about how the prediction of the centers (Y) is achieved and how the conformity levels of different interval Q are calculated.
2.	For the online update process, it seems that the process will generate a smaller alpha for a higher conformity level if the prediction always fails, which may lead to a larger prediction interval. I wonder how this process can affect the decision-making process of the RL agent. Based on my understanding, alpha will first affect the prediction interval, and further affect the d_pre in the state of the agent, while alpha seems not included in the state. Therefore, I wonder whether the RL agent can figure out whether the change in d_pre is caused by the fluctuation of the demand side request distribution or simply the change of alpha, and whether this may affect the decision results.

About Spatial-Temporal Attention Network:
1.	The time-mask operation for time dimension embedding should be described in detail. What does “The time mask here is used to hide the requests from previous time steps” mean? Whether it just filters out those requests that are fulfilled or expired, or it use some pre-defined rules to fine-tune the raw features h?
2.	For the attention network, the notations are vague. What is the definition of x^\bar? I strongly recommend that the author should also put the notations in Fig 4 to make the process clear.
3.	The RNN network includes information of previous actions (courier locations). As mentioned above, according to the Markov property, I wonder whether this is necessary for the RL agent to make future decisions. If it is necessary, why not directly put this information in the state of the agent? The design intuition of the Spatial-Temporal Attention Network should be described to tell the necessity of the components and why the current design is the best practice.
4.	How is the critic designed? Whether it have the same bottom architecture as the actor and has different output layers (e.g. MLP instead of attention)?

About Experiments:
1.	I wonder whether the model is trained and tested on the data of the same courier (or in the same region) and whether the model have good spatial generalization ability, as the model seems not consider the difference in road and the distribution of AOIs for different couriers in different regions.
2.	As mentioned in 2.2 (“During the training process, the agent takes
action (i.e., chooses a location) at each decision epoch, collectively forming a route. In contrast, during testing, the agent incrementally generates a complete route based on the current state, using a well-trained model. When a new pickup request is received, the agent recalculates the route accordingly.”), why not collectively form the route as the training setting?
3.	It is better to give more details about the experiment results. For example, I wonder how different alpha can affect the final results and expect the ablation study about the online update of alpha.
4.	More details about Conformal Prediction need to be provided.

**Reviewer Confidence:**

3: The reviewer is confident but not certain that the evaluation is correct

**Scope:**

3: The work is somewhat relevant to the Web and to the track, and is of narrow interest to a sub-community

---

### Decision · Program_Chairs · 2024-01-22

**Decision:**

Accept

**Comment:**

This paper introduces a novel route planning framework for last-mile delivery. Strengths include its relevance to a critical industry problem, and experimental validation. The reviewers commend the focus on a real-world problem. However, they also identified substantial areas for improvement, including insufficient details on the RL algorithm, unclear descriptions of conformal pickup request prediction, and ambiguous explanations of the spatio-temporal attention network. Further concerns arise regarding the paper's organization, grammar, and formatting. To enhance the impact of the paper, I would urge the authors to incorporate the responses they provided here in the final version of the paper, and improve overall presentation.